# A Luminescent Guest@MOF Nanoconfined Composite System for Solid-State Lighting

**DOI:** 10.3390/molecules26247583

**Published:** 2021-12-14

**Authors:** Tao Xiong, Yang Zhang, Nader Amin, Jin-Chong Tan

**Affiliations:** 1Multifunctional Materials & Composites (MMC) Laboratory, Department of Engineering Science, University of Oxford, Parks Road, Oxford OX1 3PJ, UK; tao.xiong@eng.ox.ac.uk (T.X.); yang.zhang2@eng.ox.ac.uk (Y.Z.); 2Department of Chemistry, University of Oxford, Mansfield Road, Oxford OX1 3TA, UK; nader.amin@chem.ox.ac.uk

**Keywords:** Luminescent Guest@MOF (LG@MOF), rhodamine B, fluorescein, nanoconfinement, quantum yield (QY), MOF-LED, guest-host interactions

## Abstract

A series of rhodamine B (RhB) encapsulated zeolitic imidazolate framework-8 (RhB@ZIF-8) composite nanomaterials with different concentrations of guest loadings have been synthesized and characterized in order to investigate their applicability to solid-state white-light-emitting diodes (WLEDs). The nanoconfinement of the rhodamine B dye (guest) in the sodalite cages of ZIF-8 (host) is supported by fluorescence spectroscopic and photodynamic lifetime data. The quantum yield (QY) of the luminescent RhB@ZIF-8 material approaches unity when the guest loading is controlled at a low level: 1 RhB guest per ~7250 cages. We show that the hybrid (luminescent guest) LG@MOF material, obtained by mechanically mixing a suitably high-QY RhB@ZIF-8 red emitter with a green-emitting fluorescein@ZIF-8 “phosphor” with a comparably high QY, could yield a stable, intensity tunable, near-white light emission under specific test conditions described. Our results demonstrate a novel LG@MOF composite system exhibiting a good combination of photophysical properties and photostability, for potential applications in WLEDs, photoswitches, bioimaging and fluorescent sensors.

## 1. Introduction

Luminescent materials are key to a variety of technologies, such as white-light-emitting diodes (WLEDs), optical sensors [1], optoelectronics [2], bioimaging [3] and so on. Among these applications, solid-state WLEDs [4,5] are of special importance as they provide basic illumination for the whole society. Two main categories of implementations to WLEDs exist: either independent red, green, and blue LEDs are arranged into arrays [6] or a blue LED is coated with a yellow phosphor [7,8,9,10]. The latter implementation, the (partial) phosphor-conversion approach, is by far more popular and successful as it overcomes many drawbacks for the former by, for example, eliminating the need for additional complicated circuitry engineering to coordinate the component LEDs. There are also efforts to develop electroluminescence-based broadband-emitting WLEDs covering the whole visible band, with some success [11]. However, more research is needed for the improvement of the performance and stability. In the phosphor-conversion approach, a growing interest is in developing phosphors free of environmentally unfriendly rare-earth elements (REEs). Although various yellow phosphors have been investigated [12,13,14,15,16,17,18,19,20,21], the market demand for cost-effective and high performance WLEDs is still increasing [22] and thus further research is needed.

In our previous work [23], we encapsulated an organic dye molecule, fluorescein, into a sodalite cage-type metal–organic framework (MOF), Zeolitic Imidazolate Framework-8 (ZIF-8), to fabricate a composite fluorescein@ZIF-8 (Fluo@ZIF-8 for short) material. MOFs are nanoporous framework materials with potential applications in such diverse areas as dielectrics and optoelectronics [24,25], sensors [26,27,28], gas storage [29], catalysis [30,31], and biomedicine [32,33]. By nanoconfinement of fluorescein guest in the ZIF-8 host, we found the quantum yield (QY) of fluorescein is increased up to 98%, due to the reduction of intermolecular interactions between the fluorescein molecules that result in aggregation-induced quenching in the solid state (NMR established a loading concentration of 1 fluorescein guest per ~3700 cages [23]). We implemented a few WLED prototype devices using these fluorescein@ZIF-8 materials, and demonstrated their tunability of color temperature. One potential drawback, however, is still not solved, i.e., a compromise has to be made between higher QY and more purity in the white color. Lower loading of fluorescein to ZIF-8 corresponds to a higher QY but more green/yellow component in the produced light.

In this work, we aim to address the foregoing challenge by investigating the photophysical properties and optical performance of the RhB@ZIF-8 system, an organic dye rhodamine B (RhB) guest confined in a ZIF-8 host in a similar fashion to the fluorescein@ZIF-8. In the literature, we found that RhB@ZIF-8 materials have been synthesized using different methods and studied for different applications such as gas adsorption [34], bioimaging [35], and water quality monitoring [36]. In particular, in the context of WLEDs, RhB has been encapsulated into ZIF-8 together with other dye molecules to produce multicomponent-dyes@MOF materials [37]. However, this approach requires special attention to controlling the precise ratio and topological distribution of the component dyes during the synthesis stage in order to achieve good lighting performance and the dyes@MOF is excited by UV instead of blue light, i.e., the full phosphor-conversion approach is adopted. In this study, we will show that by combining RhB@ZIF-8 with a low RhB guest loading (this work) and Fluo@ZIF-8 (previously reported system [23]) with a low fluorescein loading in a simple post-synthetic mixing step, both a high QY and good quality in the emitted white color can be achieved via the partial phosphor-conversion approach, therefore providing an alternative facile strategy to fabricate an efficient hybrid dye@MOF photoluminescent material for various lighting applications.

## 2. Results

### 2.1. Synthesis and Structure of RhB@ZIF-8

The RhB@ZIF-8 samples were synthesized by adopting the high concentration reaction (HCR) method [19,38,39]. Briefly, two methanolic solutions containing RhB and Zn(NO_3_)_2_ · 6H_2_O, respectively, were combined with another methanolic solution of 2-methylimidazole and triethylamine (Et_3_N) at room temperature (see Section 3). A pink to red color solid product (whose color varies depending on the concentration of the RhB used) was formed instantaneously. Fluo@ZIF-8 corresponding to a fluorescein concentration of 0.01 mM was synthesized by employing a similar methodology (in accordance with [23]).

Powder X-ray diffraction (PXRD) was used to confirm the crystalline structure of the RhB@ZIF-8 samples (Figure 1a). An agreement can be observed between the simulated pattern for ZIF-8 and the experimental one for ZIF-8 and RhB@ZIF-8 samples, respectively, confirming the formation of the crystalline host structure. Moreover, the broadening of the Bragg peaks suggests that the samples consist of nanosized crystals [40] (see atomic force microscopy (AFM) characterization below). No evidence of the RhB guest is evident from the PXRD patterns, which can be explained by the low concentration of guest used. To accurately quantify the amount of guest confined in the ZIF-8 host, we performed solution ^1^H nuclear magnetic resonance (NMR) spectroscopy to yield the RhB guest loading results summarized in Figure 1b. Encouragingly, we established a clear correlation between the amount of RhB guests applied at HCR synthesis stage (denoted as 0.01−1 mM) to the precise number of guest molecules detected from NMR. Of note, the number of cages per RhB guest determined here (7246, 1008, 186 for 0.01, 0.1, 1 mM, respectively) is appreciably lower than those established for the reported fluorescein@ZIF-8 samples (3703, 244, 30 for 0.01, 0.1, 1 mM, respectively [23]) employing an identical NMR method. The NMR results reveal the lower efficacy for the confinement of RhB within ZIF-8 compared with its fluorescein counterpart, which might be attributed to the bulkier RhB guest and its cation charge instead of anion for the latter.

The solid sample of 0.01 mM RhB@ZIF-8 was characterized by AFM (Figure 1c). The nominal width of the individual nanocrystal was found to be ~100 nm; they feature a 2-D nanoplate morphology with a width-to-height aspect ratio of ca. 10:1. We observed a similar nanocrystal morphology and dimensions for all the samples (Appendix A). The attenuated total reflectance–Fourier-transform infrared (ATR-FTIR) spectra of the synthesized RhB@ZIF-8 samples were measured and compared to those of pristine ZIF-8 (Appendix A), but no distinct band that could be associated with the guest species was detected from FTIR. This finding is expected because of the relatively low concentration of RhB used in the synthesis [23], as confirmed by the NMR data above (Figure 1b). Further discussions concerning the photophysical characterization of the samples will follow.

### 2.2. Photophysical Properties of RhB@ZIF-8

The excitation and emission spectra were measured from fluorescence spectroscopy (Figure 2a,b). The excitation peak maximum of RhB@ZIF-8 samples was detected at about 548 nm. Increasing the amount of RhB results in the rise of a shoulder band at about 505 nm and the broadening of the overall spectrum. Moreover, the emission spectrum undergoes a gradual red shift. We tentatively attribute these spectral changes to the formation of more RhB aggregates with an increase in the RhB concentration applied at HCR synthesis.

Optical band gaps of the RhB@ZIF-8 samples are estimated from the Kubelka–Munk (KM) function (Figure 2c). A weak negative correlation between the band gap and the amount of RhB is observed, which is similar to the observation in our previous work on Fluo@ZIF-8 [23]. The trend of band gap variation with RhB amount is an indicator of the effect of intermolecular RhB interactions, in agreement with our interpretation for the excitation and emission spectra.

Quantum yield (QY) of the solid-state RhB@ZIF-8 samples are 99%, 94% and 71%, respectively (Figure 2d). These are all higher than the corresponding QY values for RhB in methanol solutions with concentrations 0.01 mM, 0.1 mM and 1 mM, respectively. The QY for the 0.01 mM solution, for example, is only 49% according to our measurement. The lower QYs of the solutions are due to stronger aggregation-induced quenching than in the RhB@ZIF-8 materials, where nanoscale confinement enables the dilution of the fluorophores in the solid-state through partitioning in the periodic cages of the ZIF-8 host.

Lifetime data were obtained from photodynamic measurements based on the time-correlated single photon counting (TCSPC) technique (Figure 3, Appendix A and Appendix A). The lifetime constants in Appendix A were determined by fitting the experimental decay data shown in Appendix A. For lower amounts of RhB (0.01 mM and 0.1 mM samples), two lifetime components are observed, whereas for the largest amount (1 mM sample), three components are needed to fit the data. The contributions of *τ*_3_ and *τ*_2_ do not change much over the 0.01 mM and 0.1 mM RhB@ZIF-8 samples, but the contribution of the former significantly drops and for the latter it increases for the 1 mM RhB@ZIF-8 sample. Furthermore, the lifetime of RhB in relatively diluted methanol and ethanol solutions (<1 mM), respectively, was found to be in the range 2.3−2.8 ns, and for the more concentrated solutions (>10 mM), the values decrease below 2 ns; the lowering of lifetime in higher concentration solutions was attributed to quenching mechanisms related to aggregation [41]. Therefore, we assign the largest lifetime (*τ*_3_ = 7−8 ns) to monomers, the intermediate one (*τ*_2_ = 4−5 ns) to aggregates, and the shortest component (*τ*_1_ = 2 ns) to surface species. The *τ*_3_ component is significantly larger than the reported lifetime of RhB in dilute solutions, which we attribute to the caging effect. The *τ*_1_ component appears only for a relatively high amount of RhB (1 mM sample) because then it becomes more difficult to wash away the RhB attached to the surface of the host crystals. The lifetime analysis is consistent with the arguments made in the discussion of the excitation and emission spectra as well as the QY. The discussions above are broadly consistent with the ones for our previous work on the Fluo@ZIF-8 materials [23].

### 2.3. A Hybrid Guest@MOF Solid-State White Light Emitting Device (WLED)

Although Fluo@ZIF-8 alone can serve as a phosphor for a blue-source WLED [23], there is still a drawback for this system: a compromise has to be reached, i.e., we use either a Fluo@ZIF-8 with higher fluorescein loading to produce a near-white color while sacrificing the QY, or one with lower fluorescein loading to increase the QY at the cost of accepting some green/yellow impurity in the white color. The problem with the latter option might be alleviated by mixing in some reddish color component in the phosphor. Since the low loading (0.01 mM) RhB@ZIF-8 material possesses a comparable high QY as the 0.01 mM Fluo@ZIF-8, it will be interesting to investigate their combined performance.

We prepared a mechanical mixture (see Section 3) of the 0.01 mM Fluo@ZIF-8 and 0.01 mM RhB@ZIF-8 powder materials with a mass ratio of 1:1 (Figure 4a), and sandwiched it between two glass coverslips to form a “film” with a thickness of about 0.5 mm. The film was placed on top of a blue LED source [23] (450 nm) and the emission spectra were recorded while varying the blue light intensity by tuning the power of the LED (Figure 4b).

Within the operating power range (0.02−0.3 W) of the blue LED, both the transmitted blue light and the emitted greenish and reddish light increase in intensity as expected (Figure 4b). The contribution of the blue light increases slightly more as evidenced in the CIE color chromaticity chart (Figure 4c), but the combined white color as perceived by the naked eye makes little difference (corner inset in Figure 4c). It appears that under the conditions specified (mass ratio of the component phosphors being 1:1 and the thickness of the hybrid material being about 0.5 mm) and within the operating power range of the device, the variation of the color rendering is negligible. This is useful in scenarios where both the stability and intensity tunability of the color are desired.

To study the photostability of the 0.01 mM RhB@ZIF-8 material, the powder sample was exposed to a 450 nm blue light from the FS-5 spectrofluorometer for an extended period of 24 h with a saturated emission signal (see Appendix A). The emission maximum for the sample declines over time, but with decreasing gradient as a function of time. In total, the sample has a decrease of the emission maximum by about 13% in this accelerated photodegradation study. Overall, the 0.01 mM RhB@ZIF-8 framework is on a par with the 0.01 mM Fluo@ZIF-8 material in terms of photostability [23].

## 3. Materials and Methods

### 3.1. Chemicals and Reagents

Rhodamine B (RhB, Fisher Scientific, Hampton, NH, USA), fluorescein (Sigma-Aldrich, Saint Louis, MO, USA), methanol (MeOH, Sigma-Aldrich), zinc nitrate hexahydrate (Zn(NO_3_)_2_·6H_2_O, Fisher Scientific), 2-methylimidazole (mIm, Fisher Scientific), triethylamine (Et_3_N, Fisher Scientific) were of analytical grade and used as received without further purification.

### 3.2. Materials Synthesis

The synthesis of RhB@ZIF-8 samples followed the same procedures reported in our previous study on Fluo@ZIF-8 [23], but with fluorescein replaced by RhB. RhB in MeOH solutions (referred to as solutions A) with concentrations 0.01, 0.1, and 1 mM, respectively, were prepared. For this, 892.5 mg of Zn(NO_3_)_2_·6H_2_O was dissolved in 10 mL of MeOH, and 492.6 mg of 2-methylimidazole (mIm) and 0.837 mL of triethylamine (Et_3_N) were added to 10 mL of MeOH, to form solutions B and C, respectively. Solutions A, B and C were combined immediately in a proportion of 25 mL:10 mL:10 mL and stirred for 5 min until the reaction was completed.

The product was centrifuged at 10,000 rpm for 10 min and the liquid phase was disposed of. The solid product was washed with 40 mL of MeOH, sonicated for 5 min, and centrifuged again under the same conditions. The same procedure was repeated for a total number of 9 times; note that rigorous washing was implemented to ensure removal of RhB adhered to the external surfaces of ZIF-8 crystals. After the last centrifugation step, the liquid phase was disposed of and the solid product was dried in an oven at 90 °C for 1 h. The product was cooled to room temperature and gently ground to powder form using a mortar and pestle. The samples were labeled, as 0.01 mM RhB@ZIF-8 for example, according to the concentration of the respective solution A from which they were prepared.

For comparison, pristine ZIF-8 was also synthesized similarly, except that RhB solution (solution A) was not included in the reaction mixture, i.e., only solutions B and C were combined to form ZIF-8. A flow chart depicting the synthesis methodology of fluorescein@ZIF-8 (which applies equally to the synthesis of RhB@ZIF-8 if the fluorescein dye is replaced by RhB) can be found in the Supporting Information of our recent work [23].

### 3.3. Materials Characterization

PXRD measurements were performed on a Rigaku MiniFlex X-ray diffractometer with a Cu Kα source (1.541 Å). Fourier-transform infrared (FTIR) spectra were obtained on a Nicolet iS10 FTIR spectrometer equipped with a diamond attenuated total reflectance (ATR) cell. Atomic force microscopy (AFM) images were collected using a Neaspec s-SNOM microscope operating in the tapping mode. The guest loadings were characterized by solution ^1^H NMR spectroscopy at 298 K using a Bruker AVANCE NEO spectrometer operating at 600 MHz, equipped with a BBO cryoprobe. Further details on NMR sample preparations and analysis of spectra are described in the Supporting Information (Appendix A).

Steady-state fluorescence spectra, steady-state diffuse reflectance spectra, luminescence quantum yield (QY), and time-correlated single photon counting (TCSPC) emission decay data were recorded using the FS-5 spectrofluorometer (Edinburgh Instruments) equipped with the appropriate modules for each specific experiment. For TCSPC measurements, the samples were pumped with a 365 nm EPLED picosecond pulsed laser source. Lifetime fitting of the time constants from decay data was performed using the Fluoracle software. Corrections for instrumental function have been made for all fluorescence measurements.

The quantum yield measurement of the RhB@ZIF-8 powder samples were carried out using the SC-30 integrating sphere module of the FS-5 spectrofluorometer. The reference measurement was performed with the two holders filled with barium sulfate (BaSO_4_). The sample measurements were performed with the sample illuminated directly and indirectly, respectively, with the sample filling one holder and BaSO_4_ the other. The bandwidth for excitation and emission were 5.2 nm and 0.52 nm, respectively. The excitation wavelength used was 500 nm, and the signal was recorded from 500 to 850 nm with a step size of 0.5 nm and a dwell time of 0.2 s. The results from the reference, direct, and indirect measurements are analyzed with the Fluoracle software to calculate the quantum yield.

### 3.4. Fabrication of the WLED Device

The mechanical mixture of RhB@ZIF-8 and Fluo@ZIF-8 was prepared through a straightforward physical mixing process: by adding an arbitrary mass of the former to an equal mass of the latter, then gentling combining the two materials with a metal spatula until the color of the physical mixture appeared uniform. The uniformity of the mixture was confirmed by randomly sampling the combined materials three times and comparing the corresponding fluorescence spectra. No significant deviation was found among the emission spectra.

For MOF-LED device fabrication, a small amount (~mg) of the hybrid MOF nanoparticles was evenly spread over a thin glass coverslip and covered by another slip to create a thin film ‘sandwich’ structure with an approximate thickness of 0.5 mm. The edges of the coverslips were sealed with a tape to hold the material in place. The sealed device was then positioned on top of a 3.6 V blue LED SMD source (450 nm) for white light studies.

## 4. Conclusions

A series of RhB@ZIF-8 composite nanomaterials consisting of RhB guests nanoconfined in the porous ZIF-8 nanocrystals were synthesized and systematically studied to understand their photophysical properties. The samples were rigorously washed to minimize the guest residues adhered on the crystal surface. Electronic excitation and fluorescence spectra in combination with lifetime analysis indicate that the majority of the RhB dye molecules are encapsulated in the MOF, if the loading is not too high. These materials exhibit an extremely high QY (~99%) as the guest loading is reduced to a low concentration of 1 RhB guest molecule per ~7250 cages of ZIF-8 host (as determined by NMR). Combined with a high QY, low dye loading, green light emitting fluorescein@ZIF-8 material, the red light emitting RhB@ZIF-8 material with a comparable QY and lower dye loading constitutes a yellow phosphor which, powered by a suitable blue light source, produces stable, intensity tunable, near-white light under specific test conditions.

## Figures and Tables

**Figure 1 molecules-26-07583-f001:**
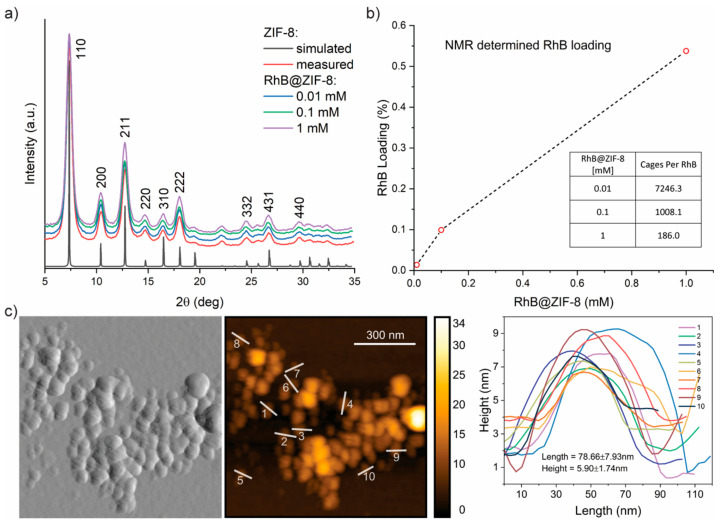
(**a**) PXRD patterns of the simulated and synthesized ZIF-8 and the three synthesized composite samples of RhB@ZIF-8 featuring three different RhB concentrations (mM) applied during material synthesis. The simulated pattern of ZIF-8 is calculated using the crystallographic information file (CIF) obtained from the Cambridge Structural Database (CCDC code: VELVOY). (**b**) RhB guest loadings determined by solution ^1^H NMR spectroscopy. The horizontal axis represents the specific guest concentration applied at material synthesis (0.01, 0.1, 1 mM). The dotted lines show the overall trend. (**c**) AFM image of the RhB@ZIF-8 nanocrystals and the corresponding height topography extracted from the designated paths on the 3D surface reconstruction of the AFM data.

**Figure 2 molecules-26-07583-f002:**
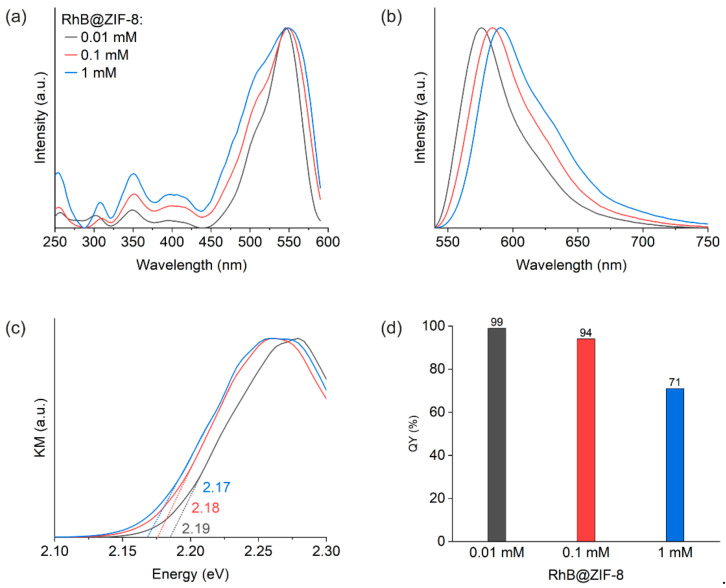
(**a**) Normalized excitation spectra (observed at emission wavelength of 610 nm). (**b**) Normalized emission spectra (observed at excitation wavelength of 520 nm). (**c**) Optical band gaps estimated from the Kubelka–Munk (KM) function for RhB@ZIF-8, employing data measured by diffuse reflectance spectroscopy. (**d**) Quantum yield (QY) of the solid-state samples.

**Figure 3 molecules-26-07583-f003:**
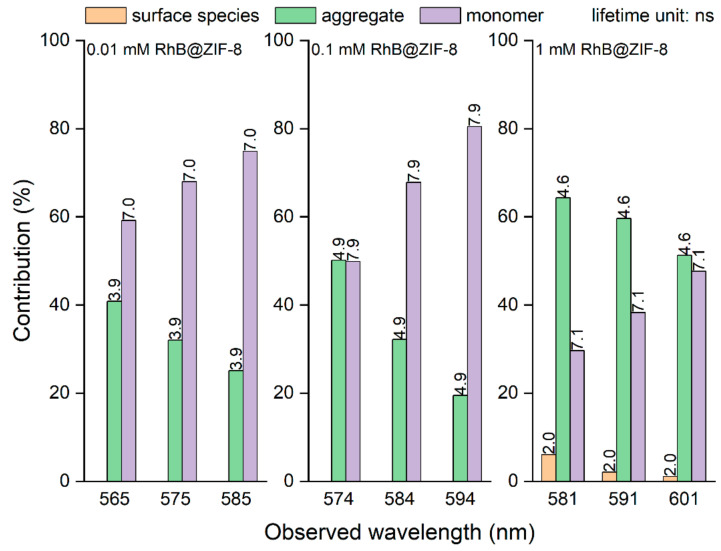
mM RhB@ZIF-8 material presents it as a natural candidate for applications where energy efficiency is crucial (Appendix A). We therefore explore the applicability of the 0.01 mM RhB@ZIF-8 material as a WLED component phosphor in the next section.

**Figure 4 molecules-26-07583-f004:**
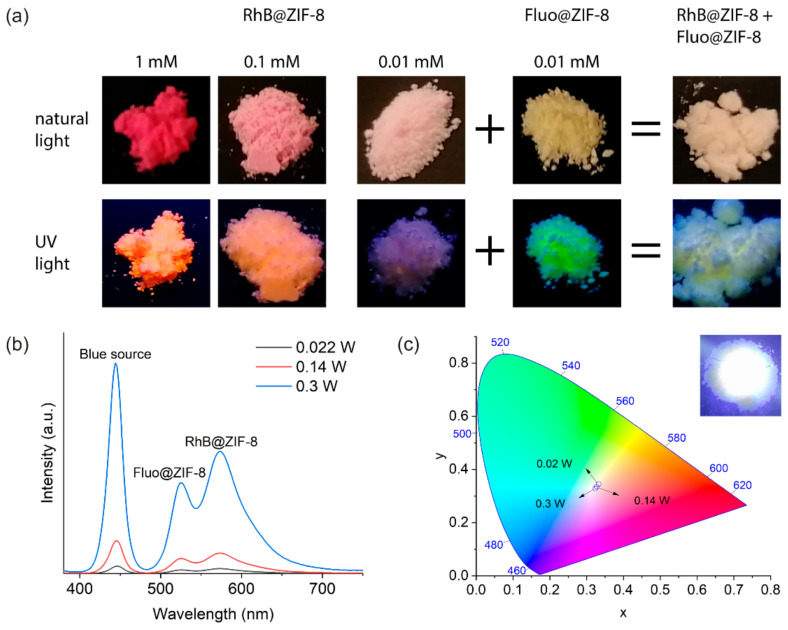
(**a**) RhB@ZIF-8 samples with different RhB guest concentrations (1, 0.1, and 0.01 mM), 0.01 mM Fluo@ZIF-8 sample, and a mechanically mixed sample of 0.01 mM RhB@ZIF-8 and 0.01 mM Fluo@ZIF-8 with a mass ratio of 1:1, viewed under natural light (**top row**) and their luminescence when excited under a 365-nm UV lamp (**bottom row**). The QY of the hybrid material was found to be 93%. (**b**) Emission spectra of a film of the mixed sample excited by a blue 450-nm LED source. (**c**) CIE 1931 chromaticity diagram of the emission color of the blue LED source coupled with the mixture film. The inset is the actual white color of the operating device to the naked eye.

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
