# Peer review of "A Luminescent Guest@MOF Nanoconfined Composite System for Solid-State Lighting"

_molecules, 2021, doi:10.3390/molecules26247583_

Round 1
Reviewer 1 Report
In my opinion, the article can be published after major revision. I would recommend expanding the list of references and a more careful approach to characterizing materials. List of specific notes below
- Powder XRD: Figure 1a:
a) Is it possible to meassure PXRD from 2 degrees and at least 50
b) Peak at ~ 22 deg on the theoretical PXRD pattern has a much lower relative intensity than on experimental.
c) The peaks at ~ 23.5 deg, ~ 28 deg, ~ 32deg, ~ 33 deg, having almost the same relative intensity in theoretical XRF but practically do not appear in experimental ones. Why?
d) It would be good to attribute Miller indices to the peaks in the PXRD pattern. Is it possible to make a precision experiment and evaluate the change in the cell parameters during the sorption of RhB, if such a change changes its place?
On the whole, XRD gives the impression of not high crystallinity of the samples and the presence of some kind of impurity phase.
2) Luminescent meassurements:
a) It is required to insert for the SI the kinetic curves in coordinates Log (I) -t together with a theoretical description of 2 or 3 exponents and the values of R2
b) What is the error in measuring the quantum yield of luminescence? Are the values 94% and 99% really different within the margin of error? Details of the quantum yield measurements should be given in Materials and Methods. Have corrections been made for instrumental function for all fluorescence measurements?
c) Page 3 line 123: "The excitation and emission spectra were measured from fluorescence spectroscopy" - is it possible to obtain excitation and emission spectra by another method?
3) Synthesis and characterisation
a) The graph in fig. 1b is not indicative, due to the points at 0 and 0.01 practically merged. It is not clear why the relationship is not completely linear.
b) It would be good to provide additional methods for the characterization of MOFs, such as DTA and BET porosimetry. This will significantly improve performance. The discussion of FTIR spectra should be extended.
4) Page 6, line 193 "redd-ish" - probaby reddish?
Reviewer 2 Report
Comments file has been attached

Reviewer 3 Report
Comment & Suggestions for Authors:
In this work, the authors reported A Luminescent Guest@MOF Nanoconfined Composite System for Solid-State Lighting which is a continued of their recently published research work (Xiong, T., Zhang, Y., Donà, L., Gutiérrez, M., Möslein, A. F., Babal, A. S., ... & Tan, J. C. (2021). Tunable Fluorescein-Encapsulated Zeolitic Imidazolate Framework-8 Nanoparticles for Solid-State Lighting. ACS Applied Nano Materials, 4(10), 10321-10333).
Here is a list of the minor issues that need to be addressed by the authors:
- Lines 16-18: the language & grammar of this sentence to be checked.
- Line 162-here reference(s) is to be given to support the sentence about the reported lifetime of RhB in dilute solutions.
- Materials * Methods section: A statement about the suppliers of the used materials such as MeOH, Et3N, Zn salt, …..etc need to be added to this section. Also, the authors are required to indicate whether the used solvents/chemicals are used as received or after certain purification/drying process.
- I suggest the authors to separate materials and methods into subsections so that the synthetic procedures of the samples are clearly indicated from other general experimental and spectroscopic methods.
Reviewer 4 Report
My review
A Luminescent Guest@MOF Nanoconfined Composite System for Solid-State Lighting
Luminescent materials find a number of applications in science, technology and everyday life. They are used in CRT monitors - cathode-ray tubes, fluorescent lamps, LEDs, optical communication amplifiers, optical memories, and are also beginning to attract attention in areas such as energy conversion, chemical detection, photovoltaics and bioimaging.
The use of luminescent materials in light emitting devices and displays is of particular interest. In many applications, semiconductor WLEDs play a role in providing basic lighting to the public.
Implementing the assumptions of the work A Luminescent Guest @ MOF Nanoconfined Composite System for Solid-State Lighting, the authors synthesized and characterized a number of composite nanomaterials of the zeolite imidazolate skeleton (RhB@ZIF8) in capsules with rhodamine B (RhB) with various concentrations of guest charges in order to investigate the possibility of their application. in semiconductor light emitting diodes (WLED).
RhB@ZIF-8 composite nanomaterials series consisting of fine-grained RhB nanocapacitors in porous ZIF-8 nanocrystals were synthesized using an appropriate procedure. The reaction was carried out in a methanol solution at room temperature.
The photophysical properties of imidazolate-8 backbone RhB@ZIF-8 were investigated, excitation and emission spectra were measured by fluorescence spectroscopy. The excitation peak maximum of RhB@ZIF-8 samples was detected at approximately 124,548 nm. It was observed that increasing the amount of RhB causes an increase in the shoulder band (by about 125-505 nm) and a broadening of the entire spectrum, respectively. Moreover, it was observed that the emission spectrum was progressively redshifted.
RhB dye molecules are encapsulated in MOF (if the load is not too high). These materials show a very high QY (~ 99%) because the visitor load is reduced to a low concentration of one RhB guest molecule in ~ 7,250 ZIF-8 host cages.
In conclusion, a series of imidazolate-8 zeolite backbone (RhB @ ZIF8) composite nanomaterials in Rhodamine B (RhB) capsules with varying guest charge concentrations were synthesized and characterized. The research on the applicability of these nanomaterials in semiconductor light emitting diodes (WLED) has been carefully conducted.
The research on the Guest @ MOF luminescent composite system has been documented with appropriate results/drawings. Modern laboratory techniques were used in the conducted works.
The work A Luminescent Guest @ MOF Nanoconfined Composite System for Solid-State Lighting is an interesting proposition of experimental aspects of Guest @ MOF luminescent composite systems.
In conclusion, I state that the literature review should be supplemented with a few current items confirming the purposefulness and importance of the research conducted. According to the reviewer, the work A Luminescent Guest @ MOF Nanoconfined Composite System for Solid-State Lighting, after text correctio/areful linguistic correction and a special emphasis on the importance of research conducted mainly in the area of luminescent materials, is suitable for printing.
Author Response
We thank the reviewer for their insightful summary of the work and for supporting the publication of this study. In the revised manuscript, the literature review section has been enhanced through the addition of recent developments in the field of WLEDs and luminescent MOFs, see our new References [1-5].
Round 2
Reviewer 1 Report
The article has been significantly improved, the revisions of the reviewers have been largely taken into account. However, a note:
"2) Luminescent meassurements:
a) It is required to insert for the SI the kinetic curves in coordinates Log (I) -t together with a theoretical description of 2 or 3 exponents and the values of R2 "
not taken into account, but seems to be important. The curve given by the authors (figure S5), apparently, refers to the process of material degradation. At the same time, it is necessary to give the luminescence decay curves, from which the tau1-tau3 times were determined (in the text, page 4, line 150 - page 5, line 169).
Author Response
Thanks for clarifying this point. In the revised version of the Supporting Information (Round 2), we have added the luminescence decay curves plotted as Log(I) vs. t, see new Figure S6. The fitted constants (tau1-tau3) are presented in Table S1, and we have added the physical description of the components and quality of the fit (chi^2).